# Gut Bacteria and Neurotransmitters

**DOI:** 10.3390/microorganisms10091838

**Published:** 2022-09-14

**Authors:** Leon M. T. Dicks

**Affiliations:** Department of Microbiology, Stellenbosch University, Private Bag X1, Matieland, Stellenbosch 7602, South Africa; lmtd@sun.ac.za

**Keywords:** gut bacteria, neurotransmitters

## Abstract

Gut bacteria play an important role in the digestion of food, immune activation, and regulation of entero-endocrine signaling pathways, but also communicate with the central nervous system (CNS) through the production of specific metabolic compounds, e.g., bile acids, short-chain fatty acids (SCFAs), glutamate (Glu), γ-aminobutyric acid (GABA), dopamine (DA), norepinephrine (NE), serotonin (5-HT) and histamine. Afferent vagus nerve (VN) fibers that transport signals from the gastro-intestinal tract (GIT) and gut microbiota to the brain are also linked to receptors in the esophagus, liver, and pancreas. In response to these stimuli, the brain sends signals back to entero-epithelial cells via efferent VN fibers. Fibers of the VN are not in direct contact with the gut wall or intestinal microbiota. Instead, signals reach the gut microbiota via 100 to 500 million neurons from the enteric nervous system (ENS) in the submucosa and myenteric plexus of the gut wall. The modulation, development, and renewal of ENS neurons are controlled by gut microbiota, especially those with the ability to produce and metabolize hormones. Signals generated by the hypothalamus reach the pituitary and adrenal glands and communicate with entero-epithelial cells via the hypothalamic pituitary adrenal axis (HPA). SCFAs produced by gut bacteria adhere to free fatty acid receptors (FFARs) on the surface of intestinal epithelial cells (IECs) and interact with neurons or enter the circulatory system. Gut bacteria alter the synthesis and degradation of neurotransmitters. This review focuses on the effect that gut bacteria have on the production of neurotransmitters and vice versa.

## 1. Introduction

Most studies on the human gut focus on the composition of the gut microbiome, variations in gut microbiota with changing diets or medication, the role bacteria play in digestion and immune responses, and regulation of entero-endocrine signaling pathways. The more we discover about the gut microbiome, gut–brain axis (GBA), hypothalamic pituitary adrenal axis (HPA), cognitive behavior and neuropsychiatric disorders, such as autism, depression, and schizophrenia, as reviewed by Dicks et al. [1], the more questions arise concerning the influence that gut bacteria have on the production of prominent neurotransmitters, such as γ-aminobutyric acid (GABA), dopamine (DA), norepinephrine (NE, also called noradrenaline, NAd), serotonin (5-HT), and histamine.

Neurotransmitters are divided into four main categories, i.e., excitatory neurotransmitters (glutamate (Glu), acetylcholine (Ach), histamine, DA, NE, and epinephrine (Epi), also known as adrenaline (Ad)), inhibitory neurotransmitters (GABA, 5-HT and DA), neuromodulators (DA, 5-HT, Ach, histamine and NE), and neurohormones released from the hypothalamus (oxytocin (Oxt) and vasopressin, also known as antidiuretic hormone (ADH)) [2]. Neurotransmitters released into the synaptic cleft between the presynaptic- and postsynaptic membranes are either destroyed by enzymes or reabsorbed into the terminal button of the presynaptic neuron by reuptake mechanisms and then recycled. Examples of fast and short-lived neurotransmitters with an excitatory action are Ach, NE and Epi. GABA is the major inhibitory neurotransmitter. As soon as a neurotransmitter binds to the receptor on the postsynaptic membrane, ligand-gated ion channels either open or close, regulating the flow of Ca^2+^, Na^+^, K^+^ and Cl^−^. Opening of the ion channels leads to a stimulatory response and closing leads to an inhibitory response [2]. Neuromodulators remain for a longer period in the synaptic cleft, modulating the activity of neurons [2]. Neurohormones are secreted into the bloodstream and transported to tissue [2].

Gut bacteria use primarily GABA, DA, NE, 5-HT and histamine to communicate with the central nervous system (CNS) [3], but also intermediate compounds, notably short-chain fatty acids (SCFAs) [4], tryptophan [5], and secondary bile acids [6]. Signals generated by these neurotransmitters and molecules are transported to the brain via afferent vagus nerve (VN) fibers. In response, the brain sends signals back to enterochromaffin cells (ECCs) and enteroendocrine cells (EECs) in the gut wall, and the mucosal immune system via efferent VN fibers [7]. Activation of the VN improves the integrity of the gut wall, reduces peripheral inflammation, and inhibits the release of pro-inflammatory cytokines [8]. Signals generated by the hypothalamus reach the pituitary and adrenal glands and communicate with EECs via the hypothalamic pituitary adrenal axis (HPA) [9]. The intricate control of entero-endocrine signaling and immune-responses keeps the gut microbiome in a balanced state. If unbalanced, the gut enters a state of dysbiosis, characterized by a drastic increase in *Enterobacteriaceae*, especially *Escherichia*, *Shigella*, *Proteus* and *Klebsiella*, and an increase in enterotoxin levels [10]. If left untreated, major gastrointestinal disorders, such as diarrhoea, ulcerative colitis (UC), Crohn’s disease, and other inflammable bowel diseases (IBDs) may develop [11,12,13]. In severe cases, elevated toxin levels may alter the functioning of the intestinal– and blood–brain barrier (BBB) that may lead to neurodegeneration [10,14,15,16].

Since gut microbiota co-evolved with humans and animals to perform neuronal functions, it is safe to assume that certain species, notably those that colonize the gastro-intestinal tract (GIT) first, play the largest role in the synthesis and degradation of neurotransmitters. From a philosophical point of view, one may ask what intestinal bacteria gain from the synthesis of neurotransmitters. Do bacteria use neurotransmitters solely to establish communication with the CNS to release molecules in the bloodstream that regulate physiological functions in the gut wall, or are there other more direct benefits? What role does gut bacteria play in the maintenance of neuronal pathways and how are signals, generated by gut bacteria, orchestrated to communicate with the CNS? To answer these questions, we need to have a better understanding of the receptors on neurons, neuro-signaling pathways, and other physiological functions of signaling molecules. Could SCFAs produced by gut bacteria play a central role in neurotransmission?

The purpose of the review is not to discuss metabolic pathways and the physiology of gut bacteria, but rather focus on the synthesis and degradation of neurotransmitters and identify prominent gut bacteria that influence neurotransmission.

## 2. Bacteria Dominates the Human Gut Microbiome

According to reports from the Human Microbiome Project (HMP) [17,18] and the METAgenomics of the Human Intestinal Tract (MetaHIT) consortium [19,20], the human gut is host to 2766 microbial species. More than 90% of the gut microbiome is represented by bacteria from the phyla Proteobacteria, Firmicutes, Actinobacteria and Bacteroidetes [20,21,22]. The majority of the gut bacteria are Firmicutes, dominated by Gram-positive *Lactobacillus* spp. and Gram-negative bacteroidetes [23]. Fusobacteria and Verrucomicrobia make up the remaining 10% of the gut microbiome [24].

The gut microbiome changes with fluctuations in hormone levels, variations in diet [25,26], stress and physiological changes brought about by the intake of drugs, especially antibiotics [27,28]. Most species in the GIT are represented by strains with unique phenotypic and genotypic characteristics [29,30,31]. Strains within species adapt as their environment changes. Strains within the same species isolated from different individuals have at least one variation in every hundred base pairs [32,33,34,35]. Species that cannot adapt or compete are replaced by novices that are able to regulate their own gene expressions, or alter their genetic composition [32,36]. Once adapted to the GIT, strains are not easily replaced [33]. *Bacteroides fragilis* adapted so well to humans that the species is represented by a single strain [37]. *Helicobacter pylori, Mycobacterium tuberculosis* [38,39,40], *Eubacterium rectale* [41] and *Prevotella copri* are host-specific and contain strains that are associated with individuals from specific geographic regions [42].

A healthy GIT is characterized by a balanced gut microbiome with a core population of beneficial microbiota. Strain-level structures are maintained by keeping genetic changes under control, as illustrated in the studies on *Escherichia coli*. Genome sequences of 24 intestinal isolates of *E. coli* ED1a, studied over a year, showed a mutation rate of only 6.9 × 10^−7^ per base [43]. This was, however, complemented by a reduction in population size [43], suggesting that external stress factors have a profound influence on the survival of bacteria in the GIT. Roodgar et al. [44] have shown that strains exposed to stress, such as antibiotics, develop resistance rapidly and change the overall composition of the gut microbiome. Although the outcome of the reports by Ghalayini et al. [43] and Roodgar et al. [44] were different, both illustrated that the frequency at which strains adapt to an ever-changing and stressful environment, such as the GIT, is unpredictable. Whatever changes takes place in the microbial population, these will affect the synthesis and degradation of neurotransmitters and, ultimately, communication with the CNS.

## 3. Wiring of the Gut Wall to the CNS

Most signals to and from the gut run through a bi-directional VN that exits the brain at the medulla oblongata and leaves the skull at the jugular foramen (Figure 1). Vagus nerves in the neck communicate with muscles of the pharynx and larynx that control swallowing and speech. Vagus nerves in the thorax downregulates heart rate. Branches of the VN leading to the GIT relax and contract smooth muscles and control secretion from glandular tissue. The celiac branch of the VN connects with the duodenum and the rest of the intestine to the distal part of the descending colon [45]. Preganglionic neurons of the VN in the medulla communicate with muscular and mucosal layers in the lamina propria and muscularis externa [46]. Sensory cells in the nodose ganglia send signals to the nucleus tractus solitarii (NTS), from where messages are sent to the locus coeruleus (LC), amygdala, thalamus, and rostral ventrolateral medulla [46].

The importance of signaling from the GIT to the brain is emphasized by the overwhelming presence of afferent fibers that outnumber efferent fibers by 9:1. Afferent fibers are also linked to receptors in the esophagus, liver, and pancreas (Figure 1). Although the VN is in contact with all layers of the gut wall, fibers do not cross the gut wall and are, thus, not in direct contact with gut microbiota [47]. Signals reach the gut microbiota via 100 to 500 million neurons in the enteric nervous system (ENS) in the submucosa and myenteric plexi of the intestinal wall (Figure 2), stretching from the esophagus to the anus [3,48,49]. Thus, although associated with the VN, the ENS in the small and large intestinal tract functions independently from the VN. This is possible due to independent sensory and motor neurons, capable of regulating muscle activity, gut wall motility, secretion of fluids, blood flow in mucosal layers and mucosal barrier functions. A decline in functions associated with the ENS often manifests as constipation, incontinence, and evacuation ailments. This is often observed in the elderly and is referred to as Hirschsprung’s disease or intestinal pseudo-obstruction [49]. Recent studies have shown that the ENS is dynamic and ever changing, maintained by a network of apoptotic and neurogenetic processes [50].

Neurons of the ENS stem from enteric neural crest cells (ENCCs) and are, by far, the largest of all nervous systems in the human body. Proliferation and migration of ENCCs are highly dependent on the glial cell line-derived nerve growth factor (GDNF) protein that is predominantly expressed by neurons in the septum, striatum, and thalamus. Expression of GDNF and other neurotrophic factors, such as neurturin (NTN), artemin (ART) and persephin (PSP), are regulated by Toll-like receptors TLR2, TLR4, TLR5 and TLR9 [51]. GDNF, NTN, ART and PSP bind to growth factor receptors (GFR)α-1, GFRα-2, GFRα-3 and GFRα-4, respectively, all tethered to the cell membrane with a glycosyl phosphatidylinositol anchor and to the transmembrane receptor tyrosine kinase (RET). The dimeric GFRα1–GDNF complex changes to the active tyrosine-phosphorylated form and sends a signal to ENCCs to express the ENS precursors and neurons required to alter the ENS during prenatal development and ensures further development of the ENS [52].

Neurons connected to the GIT have several chemical and mechanosensitive receptors that interact with hormones and regulatory peptides released from EECs, also known as Kulchitsky cells. Although these cells constitute only 1% of the epithelial cells in the GIT, they play an important role in maintaining gut homeostasis [53]. To date, 10 different types of EECs have been characterized. They act as sensory cells that coordinate changes in the secretion of chromogranin/secretogranin, 5-HT, neuropeptide Y (NPY), vasoactive intestinal peptide (VIP), cholecystokinin (CCK), somatostatin, glucagon-like peptide (GLP)-1/2, ghrelin, and substance P (SP) [54,55]. Receptors on these sensory cells are expressed by gut enteric neurons, but also vagal afferents, the brainstem and hypothalamus [56,57].

Levels of CCK, GLP-1 and PYY remain high for up to six hours after a meal. Production of CCK is stimulated by a protein-rich diet. CCK binds to specific receptors in the pancreas (CCK-A receptors), receptors in the brain (CCK-B receptors), and other receptors in the CNS. This sends a massage of fullness to the small intestine [58]. At the same time, lipolytic, proteolytic and carbolic enzymes are released from the pancreas [58]. When CCK interacts with calcineurin in the pancreas, transcription factors NFAT 1–3 are activated, which stimulate hypertrophy and the proliferation of pancreatic cells [59]. Somatostatin and pancreatic peptides prevent the release of CCK [60]. Gastrin, a hormone produced in the GIT, binds to CCK-B receptors, resulting in the release of gastric acid and mucosa production. High CCK levels increases anxiety [58]. A diet high in fats increases PYY_3–36_ production, whereas a protein-rich diet slows down the release of PYY_3–36_ for up to two hours. Bile acids interact with the G protein-coupled bile acid receptor TGR5 (GPBAR 1) and farnesoid X receptors (FXRs) on the surface of EECs [58,60].

Ghrelin, released by the stomach during fasting, enters the circulatory system and crosses the BBB. Once in the brain, ghrelin activates receptors on neurons in the arcuate nucleus, leading to increased production of NPY and agouti-related protein (AgRP) [61]. Receptors for ghrelin are located on the NPY and AgRP neurons in the hypothalamus [62]. High levels of NPY in the brain and spinal cord are secreted together with other neurotransmitters, such as GABA and glutamate [63]. NPY stimulates appetite and regulates the storage of energy in the form of fat, but also reduces anxiety, stress, and pain. NPY also regulates sleeping patterns and keeps blood pressure low [64]. AgPR stimulates the hypothalamic–pituitary–adrenocortical axis to release ACTH (adrenocorticotropin, also known as corticotropin), cortisol and prolactin (lactotropoin) [65]. Ghrelin also stimulates appetite through interaction with ghrelin receptors (GHSRs) located on neurons attached to the nodose ganglia of the VN [66].

Higher ghrelin levels are associated with elevated DA levels [67] that, in turn, send signals of satiety to the CNS [68]. A diet rich in oligofructose and inulin represses ghrelin production and increases GLP-1 production by endocrine L-cells of the intestinal epithelium [69]. GLP-1 activates GLP-1 receptors on cells in the pancreas, kidneys, and the GIT [70]. In mice with dysbiosis, activation of an enteric NO-dependent pathway resulted in the development of resistance to GLP-1 and insulin, and an increase in body mass [71,72,73]. GLP-1 is rapidly metabolized and inactivated by dipeptidyl peptidase IV and does not enter the circulatory system [74]. It would, thus, seem as if GLP-1 adheres to receptors in the GIT from where signals are transmitted to the brain via sensory neurons. The secretion of insulin stimulated by GLP-1 and inhibition of glucagon secretion keep glucose levels low during or immediately after a meal. GLP-1 also acts as an “ileal brake” by slowing down gastrointestinal motility [74]. Compounds structurally similar to GLP-1 (agonists) may, thus, be used to control type 2 diabetes. On the other hand, a decrease in the secretion of GLP-1 may lead to obesity. GLP-1 levels must be carefully controlled, as over secretion may lead to hypoglycemia. The effect of gut microbiota on ghrelin production is summarized in a study published by Schalla and Stengel [75]. According to this study, ghrelin production is stimulated by *Bacteroides* (certain species), Coriobacteriaceae, Veillonellaceae, *Prevotella*, *Bifidobacterium* (certain species), *Lactobacillus* (certain species), *Coprococcus* and *Ruminococcus*, but inhibited by some species of *Bifidobacterium*, *Streptococcus*, *Lactobacillus*, *Faecalibacterium*, *Bacteroides*, *Escherichia*, *Shigella* and *Streptococcus*, and members of Prevotellaceae. This clearly indicates that the regulation of ghrelin levels is species-specific, and more research is required to determine the specific factors involved.

*Bacteroides* produce molecules homologous to insulin, NPY and melanocyte-stimulation hormone (α-MSH) [76]. These molecules induce cross-reactions with immunoglobulins in the circulatory system that act directly against ghrelin, leptin, insulin, PYY and NPY [77]. Some strains of *Rikenellaceae* and *Clostridiaceae* produce caseinolytic protease B (ClpB) that mimics satiety experienced with increased levels of α-MSH [78]. Immunoglobulins produced by interaction with ClpB act against α-MSH and reduces its anorexigenic effects, leading to a decrease in satiety [78].

Enterochromaffin cells (ECs) controls reflexes and the secretion of gastric acid, but also produce 5-HT. Type D, G, I, K and L cells control enzymatic secretions, Mo cells initiate myoelectric migration, N cells regulate contractions, S cells (located in the small intestine) regulate acidity levels, A cells secrete ghrelin and nesfatin-1, and P cells secrete leptin. Goblet cells secrete glycosylated mucins into the lumen to form the mucus layer, which is important in maintaining intestinal barrier homeostasis. Mucin 2 (Muc2) binds to glycan receptors on dendritic cells (DCs) in the lamina propria to induce anti-inflammatory signals. Mucins also regulate the adhesion of microbial cells to the gut wall [79]. Secretion of Muc2 is regulated by intestinal microbiota and SCFAs. Intestinal trefoil factor (ITF) and resistin-like molecule-β (RELM-β), also produced by goblet cells, assist in the formation of mucosal barriers [80]. ITF regulates the formation of the tight junction proteins claudin and occludin, and controls cell apoptosis and the repair of epithelial cells. Claudin and occludin are supported by zonula occludins and actins produced by, and positioned between, epithelial cells [81]. Adheren proteins link with actin to form a cytoskeleton but are also involved in cell signaling and gene transcription regulation. Cadherin proteins bind to α-catenin to form a P120–catenin–cadherin complex that regulates cadherin formation in the plasma membrane [82]. RELM-β is more involved in altering T-helper 2 (Th2)-mediated responses. Cup cells account for 6% of epithelial cells in the ileum. The function of these cells is unknown. Tuft cells (taste chemosensory epithelial cells) secrete cytokines [80].

## 4. The Smaller Brain in Our Gut

The ENS (Figure 2), referred to as the “brain within the gut” or “second brain” [83], composed of an outer myenteric plexus and inner submucosal plexus, is structurally similar to the brain and operates on a similar “chemical platform” [84]. The modulation, development, and renewal of ENS neurons are controlled by gut microbiota, especially those with the ability to produce and metabolize hormones. Studies with germ-free (GF) mice [85,86] have shown that most enteric neurons are formed during embryogenesis and early postnatal life. A small population of neural crest cells that express Sox10 (a protein that acts as a transcriptional activator) colonizes the foregut, multiplies and then colonizes the entire bowel to form neurons and glial cells [87]. Development of enteric neurons relies on the presence of microbial cells, as shown in a study conducted on mice [88]. The authors have shown that neuroepithelial stem cells produce the protein Nestin and expresse the nuclear protein Ki67 3 to 15 days after the GIT of GF mice has been recolonized with microbiota. Nestin maintains the balance between neuronal apoptosis and neurogenesis [88,89] and protein Ki67 is associated with rRNA transcription [90]. A deficiency in any of these proteins is, thus, an indication of a decline in the health of neural stem cells (NSCs), and the inability to renew damaged cells [91]. Nestin is also expressed in the placenta [92] and is used as a marker to determine neuron health in the CNS of the unborn.

Several enzymes are involved in the maintenance of the ENS. Phosphatidylinositol 3-kinase (EC 2.7.1.137) and Akt (protein kinase B) in the PI3K/AKT signaling pathway promote neural survival and transmission, phospholipase C gamma 1 (PLC-γ1; EC 3.1.4.3) is involved in cell growth, apoptosis and transmission of neural cells, and a ras/mitogen activated protein kinase (RAS/MAPK) controls cell survival and neurogenesis [50,89]. ENS cells destroyed by apoptosis are replaced by newly formed cells [89]. However, little is known about the mechanisms that control ENS cell replenishment. A study published by Vicentini et al. [93] shed some light on this. The authors noted that the small intestinal tracts of antibiotic (Abx)-induced mice (thus without gut microbiota) were longer than normal mice, which led to slower transit of gut contents, increased carbachol (carbamylcholine)-stimulated ion secretion, and increased gut wall permeability. Since carbachol binds to, and activates, acetylcholine (Ach) receptors [94], increased levels would stimulate muscarinic and nicotinic receptors. Muscarinic Ach receptors form G protein-coupled receptor complexes in the cell membranes of, for instance, the parasympathetic nervous system [94]. Nicotinic acetylcholine receptors are polypeptides that are present in the central and peripheral nervous system, muscle, and many other tissues [95]. The authors also noted a decrease in neurons of the submucosal and myenteric plexuses of the ileum and proximal colon, and thus the neuronal network of the ENS. In addition, the myenteric plexus of the ileum contained fewer glial cells (neuroglia), which is an indication that neurons of the ENS had been deprived of nutrient and oxygen supply and neuronal cells were not insulated from each other, lost the ability to destroy pathogens and were unable to remove dead neurons; all of which are functions of neuroglia [96].

The role gut microbiota plays in association with neurotransmitters such as Ach and regulatory neuropeptides has become more apparent in research conducted on experimental animals. Ach, the main neurotransmitter of the parasympathetic nervous system involved in the contraction of smooth muscles, dilation of blood vessels, secretion of bodily fluids and the downregulation of heart rate, is stored in vesicles at the terminal of Ach-producing neurons. Recent findings [97,98] have shown that the secretion of Ach may be stimulated by certain species of *Lactobacillus*. Studies with animal models have shown that a stroke, created by restricting blood flow at the proximal cerebral artery, altered the composition of gut microbiota, which led to a decrease in the production of Ach, changes in peristalsis, increased gut permeability and dysbiosis [99,100]. A decline in Ach also led to an increase in adrenergic signaling and inflammation of the GIT. This, in turn, restricted the middle cerebral artery that led to a decline in goblet cells in the cecum and lowering of mucin production [101]. This indirect effect on gut health by a neurotransmitter of the parasympathetic nervous system, under microbial control, illustrates the complexity of the mechanisms involved in controlling ENS functions.

When GF mice were reconstituted with intestinal microbiota and administered lipopolysaccharides (LPS) and SCFAs, intestinal functions were restored, and enteric neurons and neuroglia recovered. Treatment with LPS assisted in the recovery of damaged neurons and gut microbiota but did not stimulate the formation of new neurons. Treatment with SCFAs, on the other hand, restored neuronal loss [93]. Concluded from the data presented in this study, a lack in SCFAs, as would be expected in patients with dysbiosis, may lead to a loss of enteric neurons, and thus a weakened ENS. This emphasizes the importance of SCFA-producing gut microbiota.

## 5. Neurotransmitters and Neuropeptides

Neurotransmitters are divided into amino acids (e.g., Glu, aspartate, D-serine, GABA and glycine), monoamines (DA, NE, Epi or Ad, histamine and 5-HT), trace amines (e.g., phenethylamine, *N*-methylphenethylamine, tyramine, 3-iodothyronamine, octopamine, tryptamine), peptides (oxytocin, somatostatin, substance P, cocaine and opioid peptides), gasotransmitters (nitric oxide, carbon monoxide and hydrogen sulfide), purines (adenosine triphosphate and adenosine), and smaller compounds, such as Ach and anandamide. Of these, Glu, GABA, glycine, DA, NE, 5-HT and histamine are considered the key neurotransmitters and will be discussed in more depth.

### 5.1. Glutamate (Glu)

Glu is the principal excitatory neurotransmitter in the brain and plays a key role in memory storage [102]. Glu is released from presynaptic nerve terminals and adheres to ionotropic glutamate receptors (iGluRs) located on postsynaptic terminals. This increases the transfer of Ca^2+^ through voltage-activated calcium channels (VACCs) located in the terminals [103] and activates calcium calmodulin-dependent kinase (CaMK; EC 2.7.11.17), extracellular signal-regulated kinase (ERK, EC 2.7.11.24) and cyclic AMP response element binding (CREB) protein, all crucial for protein synthesis and maintenance of synaptic density. This process is carefully controlled. Excess Glu is taken up by glial cells via transporters EAAT1 and EAAT2. Glu is then converted to glutamine, which is transported back to presynaptic nerve terminals and converted to Glu by glutaminase (EC 3.5.1.2). Vesicular glutamate transporters VGLUT1 and VGLUT2 then transport the newly formed glutamate to vesicles in the presynaptic neuron. The over-excitement of Glu (glutamate excitotoxicity) accelerates the progression of Alzheimer’s disease [104]. For more information on glutamate excitotoxicity, the reader is referred to the review by Esposito et al. [105].

D-Glu is part of the peptidoglycan structure in the cell walls of bacteria and is produced via Glu racemase (EC 5.1.1.3) Mur1 [106]. *Corynebacterium glutamicum, Lactobacillus plantarum*, *Lactococcus lactis*, *Lactobacillus paracasei, Brevibacterium avium*, *Mycobacterium smegmatis, Bacillus subtilis* and *Brevibacterium lactofermentum* convert L-Glu to D-Glu [20,107,108]. The latter is converted to GABA by Glu decarboxylase (GAD; EC 4.1.1.15) [109]. The metabolism of D-amino acids in the brain is regulated by gut bacteria, as shown in the studies on specific pathogen-free (SPF) mice [110]. The authors have shown that L-arginine, L-glutamine, L-isoleucine, and L-leucine were significantly higher in these animals. D-Aspartate, D-serine, and L-serine were, however, higher in certain areas of the brain. Low plasma D-Glu levels are associated with cognitive impairment in Alzheimer’s disease [111]. *Lactobacillus rhamnosus* JB-1 changed the expression of GABA receptors (GABARs) in the brain, which led to lower anxiety and less depressive behavior [111].

### 5.2. Gamma-Aminobutyric Acid (GABA)

GABA is found in high (millimolar) concentrations in many brain regions and is released into the synaptic cleft upon depolarization of presynaptic neurons [112]. GABA is also produced in glia (astrocytes) connected with pre- and post-synaptic neurons [113] but different metabolic pathways [114]. In GABA-expressing neurons, α-ketoglutarate, synthesized in the Krebs cycle, is converted to L-glutamic acid by GABA α-oxoglutarate transaminase (GABA-T, EC 2.6.1.19) and decarboxylated to GABA by GAD. In glial cells, glutamate is not decarboxylated to GABA, as these cells do not express GAD. Instead, GABA is synthesized from N-acetylputrescine with monoamine oxidase B (MAOB, EC 1.4.3.4) as a key enzyme [114]. GABA in glia is converted to succinic semialdehyde by GABA-T but also to glutamine, which is deaminated to glutamate before it re-enters the GABA shunt [112]. Several reports of GABA produced by gut microbiota have been published, with special reference to *Lactobacillus, Bifidobacterium* and *Bacteroides,* specifically *B. fragilis* [115]. *Escherichia coli* K12 uses GABA as its sole carbon and nitrogen source [116]. More recently, a new “GABA-eating” species, *Evtepia gabavorous* within the family *Ruminococcaceae*, has been described [115,117]. The dependency of *E. gabavorous* on GABA is evident in that strains only grow in the presence of GABA-producing *Bacteroides fragilis* [118].

During early life, GABA serve as neurotransmitters [119,120]. With further development, GABAergic neurons transfer glutamate between synaptic cells. The adhesion of GABA to GABARs to postsynaptic neurons prevents the transfer of Na^+^, K^+^, Ca^2+^ and Cl^−^ [121]. Three classes of GABARs have been described, i.e., GABAR_A_, GABAR_B_ and GABAR_C_. GABAR_B_ transfers signals received from hormones, neurotransmitters and pheromones to signal-transferring pathways [112,120,121]. A select few microorganisms have been known to alter the function of GABARs. *Lactobacillus rhamnosus* JB-1 changed GABARs expression in the brain, which led to a decrease in anxiety and depression [122]. It may, thus, well be that certain *Lactobacillus* spp. play a key role in the regulation of anxiety and depression. Treatment with *Lactobacillus* elevated GABA levels in hippocampal and prefrontal cortex [123]. Acetate produced by microbiota in the colon is transferred across the BBB to the hypothalamus and enters GABA neuroglial cycling pathways [124]. Gut-derived GABA, unlike catecholamines, reaches the CNS via specific GABA transporters expressed in the BBB [125]. At least six different GABA transporters regulate uptake into presynaptic nerve terminals and surrounding glial cells [126]. Re-uptake of GABA by neurons coincides with a decrease in Na^+^ levels. Under normal physiological conditions, intracellular levels of GABA are approximately 200 times higher than extracellular levels. In glia, GABA is transformed to glutamine. The latter is then transferred back to neurons [126]. Glutaminase converts glutamine to glutamate, which re-enters the GABA shunt. Other microbiota, e.g., *Akkermansia muciniphila*, *Parabacteroides merdae* and *Parabacteroides distasonis,* may also play a role in the regulation of GABA by altering GABA/glutamate ratios and increase glutamate levels in the brain [127].

### 5.3. Glycine

Glycine is an excitatory and inhibitory neurotransmitter [128]. As an excitatory neurotransmitter, it serves as a co-agonist with Glu at the N-methyl-D-aspartate receptor (NMDAR) transmitter, allowing the removal of magnesium from the passage of Na^+^ and Ca^2+^, which is critical in the enhancement of learning and neuronal flexibility [129]. As an inhibitory neurotransmitter, glycine plays a role in the processing of motor and sensory information that permits movement, vision, and audition [129]. Glycine is often co-released with GABA and moderates excitatory neurotransmission by enhancing the action of glutamate at NMDARs. Excess glycine is taken up by the sodium-and-chloride-coupled transporters GLYT1 (located in the plasma membrane of glial cells) and GLYT2 (found in pre-synaptic terminals) [130].

### 5.4. Dopamine (DA)

DA is produced in the substantia nigra, ventral tegmental area, and hypothalamus and is released into the nucleus accumbent and prefrontal cortex of the brain. It is generally referred to as the reward neurotransmitter, but also has a role in the modulation of behavior and cognition, voluntary movement, motivation, inhibition of prolactin production, sleep, dreaming, mood, attention, working memory and learning [131]. Tyrosine is hydroxylated to L-dihydroxyphenylalanine (L-DOPA) and then decarboxylated to DA. In the presence of DA, β-hydroxylase DA is converted to NE and Epi (Ad) [132]. DA is also produced by some *Bacillus* and *Serratia* species in the GIT [133].

Gut microbiota may alter the function of a neurotransmitter, as shown by studies conducted on individuals with Parkinson’s disease. Levodopa, a natural precursor of DA, crosses the blood–brain barrier when administered peripherally and increases DA levels in the brain. However, levodopa metabolized by gut microbiota lowers its availability and DA produced peripherally causes unwanted side effects. *Enterococcus faecalis* decarboxylates L-DOPA to DA, but the latter is immediately dehydroxylated to m-tyramine by *Eggerthella lenta* [134]. The conversion of L-DOPA to DA by *E. lenta* is encoded by a single-nucleotide polymorphism in the gene encoding DA dehydroxylase. In future, screening of gut microbiota or mapping of the gut microbiome may become important in selecting a drug used for psychiatric treatment. This is possible with recent advances in sequencing, cloning, genetic manipulation, viral (including bacteriophage) targeting, imaging techniques and knowledge gained through research on GF animals. This may lead to the discovery of new pathways in the GBA that regulate brain functions and behavior. With recent progress in research on sensory neurons, intestinal epithelial cells and compounds affecting the activity of neurotransmitters, future psychiatric treatment may lean towards the production of specific metabolites in fermented foods.

### 5.5. Norepinephrine (NE) or Noradrenaline (NAd)

NE is structurally similar to EPi (Ad) and is produced during excitement, but is also involved in behavior and cognition, such as memory, learning, and attention [135], and is involved in inflammation and modulates responses of the autonomic nervous system [136]. NE shares characteristics with autoinducer-3, a quorum sensing molecule that stimulates enterohemorrhagic *E. coli* motility and virulence [137,138]. Cell growth of *Klebsiella pneumoniae*, *Shigella sonnei*, *Pseudomonas aeruginosa*, *Enterobacter cloacae*, and *Staphylococcus aureus* was also stimulated by NE, most likely due to iron acquisition [139]. *Bacillus mycoides*, *Bacillus subtilis*, *E. coli K12*, *Proteus vulgaris* and *Serratia marcescens* produce between 0.45 and 2.13 mM NE [140] and there is now evidence that these species may use the hormone in quorum sensing [3,141]. GF mice had reduced levels of NE in the cecum and in tissue cells, but these could be restored by colonization with a combination of *Clostridium* spp. [133].

### 5.6. Serotonin (5-HT)

5-HT regulates appetite, gut motility, mood, cognition and sleeping patterns [142,143,144]. Enteric 5-HT levels are regulated by tryptophan hydroxylase TPH1 and TPH2 [145]. Although 5-HT is synthesized by neurons of the ENS, more than 90% of 5-HT is produced in the gut by ECCs [146]. Some authors claim that as much as 80% 5-HT is produced in the GIT by *E. coli*, *Hafnia, Bacteroides*, *Streptococcus, Bifidobacterium*, *Lactococcus, Lactobacillus, Morganella*, *Klebsiella, Propionibacterium*, *Eubacterium*, *Roseburia* and *Prevotella* [142,143]. *Candida* and *Escherichia* convert tryptophan in food to 5-HT [58]. Serotonin production by gut microbiota may have a greater effect on the CNS than originally anticipated, as ECs interact with 5-HT-receptive afferent fibers in vagal or dorsal root neurons [147]. Proof of microbiota affecting emotional behavior was shown in various studies on mice. Mice that had a subdiaphragmatic vagotomy showed no changes in emotional behavior, irrespective of an increase in neurotransmitters produced by gut microbiota [148,149].

Studies conducted on mice have shown a drastic increase in the development of enteric neurons two to three weeks after treatment with a serotonin 5-HT4 agonist [150]. The authors have also shown that neurons of GF mice, unable to synthesize serotonin, were less developed. A separate study [151] has shown that neuronal dysfunctions in GF mice could be reversed by recolonization with gut microbiota. Several studies confirmed this and presented clear evidence that the synthesis of 5-HT is regulated by gut microbiota [88,146,150]. Reigstad et al. [152] have shown that sodium acetate (10–50 mM) significantly increased *TPH1* mRNA expression in a human-derived EC cell model (BON cells). Butyrate (0.5 and 1.0 mM) increased *TPH1* mRNA expression to similar levels (more than 3-fold). However, treatment with 2.0 mM butyrate did not significantly alter TPH1 expression. Higher levels of butyrate (8.0 and 16.0 mM) suppressed TPH1 expression 13.5- and 15.7-fold, respectively, which was below the expression levels recorded for untreated EC cells [152]. These findings clearly indicated that EC cells are stimulated by gut microbiota, specifically SCFAs, to produce 5-HT. Regulation of TPH1 expression is viewed as a rate-limiting step in the biosynthesis of DA, NAd and adrenaline [153]. Changes in 5-HT levels are also associated with IBD. In patients with ulcerative colitis (UC) and Crohn’s disease (CD), a drastic increase in serotonin-immunoreactive cells was recorded in the colon [154]. In patients diagnosed with CD, colonic PYY, pancreatic polypeptide (PP), and oxyntomodulin-producing endocrine cells were much lower [154]. Patients with IBD have high levels of plasma chromogranin-A (CgA) [155], whereas patients with UC have high levels of fecal CgA [156]. CgA and its derived peptides, e.g., vasostatin (VS), catestatin (CST) and chromofungin (CHR), assist in the regulation of antimicrobial activity, suggesting that changes in CgA levels in EECs may lead to alterations in intestinal microbial composition and diversity [157]. Significant changes have also been recorded in other circulating EEC secretory products, such as PYY, CCK, GLP-1, 5-HT, somatostatin, gastrin, and motilin in individuals with IBD [158]. One of the consequences of chronic colitis is EC hyperplasia. In studies conducted on mice with colitis, an increase in 5-HT production was observed [159,160]. With an increase in 5-HT levels, expression of the serotonin type 7 (5-HT7) receptor on DCs is activated and a pro-inflammatory immune response is triggered [161]. Inhibition of the 5-HT7 receptor reduced intestinal inflammation [162]. Activation of 5-HT4 receptors plays a major role in maturation of the adult nervous system in that it regulates the formation of neurons and also protect the cells [88,150].

In germ-free mice, low levels of tyrosine led to reduced 5-HT levels [88]. A lack in communication between gut microbiota and the ENS is directly linked to dysbiosis and gastrointestinal disorders, as shown by several studies [88,163,164,165]. In a healthy gut, *Clostridium perfringens* modulates 5-HT synthesis by using TPH of the host [146].

5-HT also affects the host’s immune system and behavior of glial cells in the ENS and CNS [166,167], whilst activation of the 5-HT4 receptor in the ENS displays neurogenerative and neuroprotective properties [168]. Gut microbiota convert tryptophan to indole-3-acetic by using tryptophan monooxygenase (EC 1.13.12.3) and indole-3-acetamide hydrolase (EC 3.5.1.4), then decarboxylate indole-3-acetic to 3-methyl indole. Indole-containing compounds activate the aryl hydrocarbon receptor (AHR) on mucosal and immune cells [169,170]. The conversion of tryptophan to indole by *Lactobacillus reuteri, Lactobacillus johnsonii,* and *Lactobacillus murinus* helps in the differentiation of T cells [171] and prevents colitis [172]. Indole-3-aldehyde produced by *Lactobacillus* spp. induces AHR, which, in turn, induces IL-22 required for the secretion of antimicrobial peptides [169]. Since inflammation plays an important role in the development of Parkinson’s disease, multiple sclerosis, amyotrophic lateral sclerosis, and Alzheimer’s disease [173], production of indole by gut microbiota need to be investigated in more depth.

Another surprising finding is that 5-HT promotes the colonization of *Turicibacter sanguinis* in the human gut, a bacterium that expresses a neurotransmitter sodium symporter-related protein structurally similar to the mammalian serotonin reuptake transporter (SERT) [174]. A recent report [172] suggests that some neurotransmitters may serve as growth substrates for intestinal bacteria. High levels of 5-HT may decrease gut wall permeability, whilst low levels of 5-HT decrease the expression of occludin and weaken the gut wall, leading to increased permeability and the development of a leaky gut [143]. The latter was reported in patients diagnosed with irritable bowel syndrome (IBS) [143]. Excess 5-HT in the circulatory system is transported across the cell membrane by SERT and intracellularly inactivated by MAO [145]. Most of the currently available antidepressants prevent the synaptic re-uptake of biogenic amines [175]. On the other hand, patients with major depressive disorder (MDD) often develop resistance to antidepressants. This led to studies that investigated the relationship between gut microbiota and depression [176].

### 5.7. Histamine

Histamine facilitates homeostatic functions, endorses wakefulness, and controls feeding and motivational behavior [177]. Histamine produced by gut microbiota activates histamine receptors [178]. Bacteria that produce histamine include *Lactobacillus* spp., *Lactococcus lactis*, *Oenococcus oeni*, *Pediococcus parvulus*, *Streptococcus thermophilus*, *Morganella morganii*, *Klebsiella pneumoniae*, *Enterobacter* spp., *Citrobacter freundii* and *Hafnia alvei* [3]. Cadaverine, putrescine and agmatine also activate histamine receptors [179]. Further research is required to identify neuron-signaling microbial metabolites. It is also important to determine if these metabolites are produced at high enough concentrations, and in active form, to communicate with receptors on neurons. An area that has up to now been ignored is the role viruses and bacteriophages play in neurotransmission. Bacteriophages can alter the levels of tryptamine and tyramine in the GIT [180], but we do not know if these changes have an impact on neuronal activity.

## 6. Role of Short Chain Fatty Acids in Neurotransmission

SCFAs, such as butyrate, acetate, lactate and propionate, are largely produced in the colon by *Bifodobacterium, Lactobacillus*, *Lachnospiraceae*, *Blautia*, *Coprococcus*, *Roseburia* and *Faecalibacterium* and provide energy to epithelial cells [181]. These SCFAs adhere to free fatty acid receptors (FFARs), e.g., GPR43(FFAR2) and GPR41 (FFAR3) located on the surface of IECs [182]. Receptors FFAR2 and FFAR3 are also expressed in the ENS, portal nerve and sensory ganglia [88]. GPR 41 in the ENS transfers signals induced by SCFAs directly to the CNS [183]. GPR43, expressed in white adipose tissue, communicates with SCFAs to stimulate energy expenditure in skeletal muscles and the liver [184]. SCFAs stimulate antimicrobial peptides through the cathelicidin LL-37 pathway and prevent Shigella infection [185]. Individuals with IBD have low levels of fecal SCFAs, accompanied by a decrease in Firmicutes and Bacteroidetes [186,187].

The crossing of IECs is facilitated by specialized monocarboxylate transporters [188,189]. Some SCFAs, however, diffuse across IEC membranes and enter the circulatory system un-ionized [190]. Butyrate protects intestinal barrier function by up-regulating the tight junction protein claudin-1 [191] and is used by colonocytes as their main energy source [192]. Butyrate also induces apoptosis of colon cancer cells [192] and plays an essential role in the consumption of oxygen in epithelial cells. A balanced state of oxygen prevents dysbiosis [193]. Butyrate suppresses inflammatory responses by downregulating histone deacetylase EC 3.5.1.98 (also referred to as lysine deacetylase) inhibitors (HDACi) [194,195]. An increase in de-acetylated histones (due to the inhibition of HDACi), together with a decline in gene transcriptions, leads to autophagic cell death, the activation of extrinsic and/or intrinsic apoptotic pathways, an increase in the production of reactive oxygen species (ROS), and a decrease in the expression of pattern recognition receptors, kinases, transcription regulators, cytokines, chemokines, and growth factors [196,197]. Molecules released from dying cells, interpreted as damage-associated molecular patterns (DAMPs) and pathogens recorded as pathogen-associated molecular patterns (PAMPs), are recognized by nucleotide-binding oligomerization domain (NOD)-like receptors (NLRs) and form specific protein–protein interactions. These interactions, also prevalent in lymphocytes, macrophages and dendritic cells, play a key role in the regulation of cytokines, chemokines and the expression of genes coding for the production of antimicrobial compounds, collectively referred to as the innate immune response [183,198,199]. Downregulation of NLRs, thus, prevents the formation of multi-protein inflammasomes, the signaling of caspase-independent nuclear factor kappa B (NF-κB) and mitogen-activated protein kinase (MAPK, EC 2.7.11.24). These cascades of events counteract autoimmune and inflammatory disorders and as recently shown, repress the growth of cancerous cells [200].

Changes in butyrate levels may be due to an imbalance in butyrate-producing bacteria, or increased binding of butyrate to FFARs located on EECs [9,201]. The activation of G-protein-coupled receptors (GPCRs) by butyrate may cause several neurodegenerative disorders [9] and stimulation of regulatory T cells by butyrate induces the production of inflammatory cytokines [202]. Higher cytokine levels control the proliferation of *Proteobacteria* in the GIT [203]. Low levels of butyrate inhibit GPCRs and interrupt immune or hormonal responses [204]. Butyrate modifies the integrity of the BBB, which affects the CNS and maturation of microglia [204]. In germ-free (GF) mice, defective microglia could be stimulated by incorporating additional butyrate, propionate and acetate to the feed [205]. Acetate crosses the BBB and accumulates in the hypothalamus [124]. This activates the hypothalamic–pituitary–adrenal (HPA) axis, which sends signals to EECs [9].

Inhibition of HDACi in the frontal cortex and hippocampus of mice, brought about with the administration of sodium butyrate, alleviated depressive behavior [206], dementia and brain trauma [201]. High levels of HDAC have been reported in patients suffering from neurological disorders such as depression, Parkinson’s disease, and schizophrenia [201]. Elevated levels of ACHs, however, increase the expression of *bdnf*, encoding brain-derived neurotrophic factor (BDNF) in the frontal cortex and hippocampus and stimulate brain development [207,208]. Low levels of BDNF are associated with depression and anxiety [144,209,210]. Neurological disorders may, thus, be prevented by keeping SCFAs and HDAC at optimal levels. One way of achieving this is to maintain a well-balanced gut microbiome. SCFAs, tryptophan precursors, and metabolites interact with receptors located on the gut wall, muscle layers surrounding the gut, liver, pancreas, adipose tissue and immune cells [58].

Subsequent studies have shown that SCFAs can modulate genes encoding the cAMP response element-binding (CREB) protein that regulates the synthesis of catecholamine neurotransmitters, such as DA [211,212]. SCFAs activated the expression of tyrosine hydroxylase, the first reaction in the production of DA, and decreased the expression of dopamine- β-hydroxylase (DBH; EC 1.14.17.1), and thus the conversion of DA to NE [213,214]. These studies clearly indicated that SCFA-producing gut microbiota play a crucial role in brain processes. Further evidence for this was found in a more recent study [4] on the role that SCFAs play in alleviating stress-induced symptoms. The hippocampus and striatum are especially susceptible to DA, which triggers a “neurotransmitter reward” response, and thus a change in mood. A diet that increases the production of SCFAs may alleviate immune and metabolic dysfunctions, as shown in studies conducted on schizophrenic patients [215]. Despite the many positive attributes DA may have, the rate at which it is synthesized, including its conversion to NE, needs to be tightly regulated. A deficiency in DBH (thus elevated DA levels) may have a serious negative affect on the autonomic nervous system (ANS) that controls the regulation of blood pressure and body temperature [216]. Early symptoms of DBH deficiency include vomiting, dehydration, decreased blood pressure (hypotension), difficulty maintaining body temperature, low blood sugar (hypoglycemia) and extreme fatigue during exercise. In males, DBH deficiency (thus increased dopamine levels) may result in retrograde ejaculation, i.e., discharge of semen into the bladder [216].

Acetate reduces appetite by stimulating the secretion of ghrelin [124]. Consistent with this, propionate feeding induces fos (fos proto-oncogene, AP-1 transcription factor subunit) expression in the dorsal vagal complex of the brainstem, the hypothalamus, and the spinal cord [190], raising the question as to whether SCFA-induced stimulation of peripheral sensory neuronal activity could mediate the effects of SCFAs on host feeding behavior. SCFAs regulate several other physiological functions in the body, e.g., the maturation and functioning of microglia in the CNS [203], the transmission of signals generated by serotonin, GABA and DA signals to neurons [217,218] and the secretion of anions in the colon [192,219,220]. The latter is due to the stimulation of nicotinic Ach receptors in the colon that leads to an increase in Ach production and the stimulation of goblet cells to secrete more mucus [221].

In immune cells, SCFAs regulate the differentiation of T-cells [222,223]. In enteroendocrine cells, SCFAs stimulate the release of gut hormones [224]. Bile salts, combined with SCFAs, play an integral part in enterohepatic circulation, but are equally important in the regulation of neuronal pathways and signaling to the CNS [225]. Further research in this area may shed more light on the effects diets have on mood changes.

It was reported that individuals with Parkinson’s disease (PD) had increased cells numbers of *Enterobacteria*, but fewer *Prevotella* spp. [226,227]. Other PD patients had fewer butyrate-producing species, such as *Blautia*, *Coprococcus*, and *Roseburia*, but higher numbers of potentially harmful proinflammatory Proteobacteria, especially *Ralstonia* [228]. These changes in composition of gut microbiota coincided with lower levels of SCFAs [227].

From all these studies, it is clear that SCFAs, produced by microorganisms, play a key role in microbiota–gut–brain axis communication, protection of the intestinal barrier and inflammatory responses. Levels of SCFAs, however, need to be carefully controlled, as several disadvantages have been reported. Acetate, for instance, promotes the production of intestinal IgA [229], stimulates the secretion of cytokine IL-6 and increases neutrophil recruitment [184]. Propionate administered to patients with obesity enhanced gut hormone secretion, while reducing adiposity and overall weight gain [230]. The mechanisms involved in the regulation of SCFA production and its impact on feeding behavior remain unclear. The exact mechanisms whereby SCFAs regulate appetite remains unclear. FFAR2 and FFAR3 are expressed in the ENS, portal nerve and sensory ganglia [192,231]. This suggests that SCFAs regulate the nervous system, a hypothesis supported by the induction of fos in the dorsal vagal complex of the brainstem, the hypothalamus, and spinal cord [192]. The expression of fos is induced by serum, growth factors, tumor promoters and cytokines. SCFAs are fundamental molecules involved in regulating energy homeostasis, and SCFARs are expressed by a wide variety of non-neuronal cell subtypes as well. In immune cells, for example, SCFAs can regulate T regulatory cell differentiation [182,222] and microglial maturation [223], whereas in EECs, SCFAs can stimulate the release of gut hormones [224].

Expression of *Tph1* is induced by SCFAs [142,145]. Alleviation of anxiety and depression is, thus, associated with an increase in SCFA levels [143]. The opposite is also true. Individuals suffering from anxiety and depression have low SCFA but may also have high blood pressure—all of which may lead to cardiovascular diseases, strokes, obesity and diabetes mellitus [143]. Studies conducted on rats have shown that hypertension could be prevented by restoring acetate levels in the cecum [88,232]. These studies have shown a strong interaction between gut microbiota and the ENS, which may provide answers to the link between microbial dysbiosis and gastrointestinal disorders.

SCFAs produced by gut microbiota are transported via blood vessels to the brain and modulate functions of neurons, microglia and astrocytes, and affect the BBB [233]. Histone acetylation is regulated by SCFAs such as acetate and butyrate. It is, thus, important to control the production of SCFAs. A separate study has shown that acetate, produced in the colon by gut microbiota, crosses the BBB and concentrates in the hypothalamus, from where it stimulates GABA production in the brain [126]. Further research is required to understand how bacteria regulate neurotransmitters and lead to a sudden increase in pathogens.

## 7. Conclusions

Gut bacteria are not only sensitive to physiological variations in the GIT, but also to the signals received from the CNS via the VN and ENS. Minor activation of the VN results in drastic changes in the production of neurotransmitters, which affects digestion, intestinal permeability, gastric motility, and immune regulation. Signals received from the CNS and ENS change the microbial composition in the GIT and may benefit the survival and proliferation of certain species. In return, gut bacteria stimulate EECs to produce hormones such as 5-HT, CCK and PYY that communicate with the CNS via neural afferent fibers. Neurotransmitters such as Glu, GABA, DA, NE, 5-HT and histamine synthesized by gut bacteria communicate with the CNS, autonomic sympathetic and parasympathetic nervous systems and the HPA to control the release of growth-stimulating hormones. Intermediate compounds such as SCFAs, tryptophan and secondary bile acids also communicate with the CNS. Signals generated by the hypothalamus reach the pituitary and adrenal glands and communicate with EECs via the HPA. The intricate control of entero-endocrine signaling and immune responses keeps the gut microbiome in a balanced state. In future, stimulation of the VN by probiotic bacteria may be used in the treatment of neurological disorders, such as depression and anxiety, IBD and IBS.

## Figures and Tables

**Figure 1 microorganisms-10-01838-f001:**
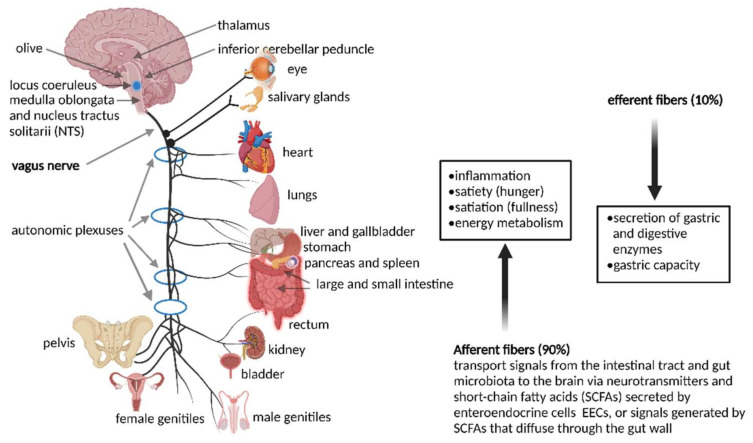
The vagus nerve (VN) connects the gastro-intestinal tract (GIT) with the central nervous system (CNS), but is also connected to various other organs. This illustration was constructed using BioRender (https://biorender.com/, assessed on 15 July 2022).

**Figure 2 microorganisms-10-01838-f002:**
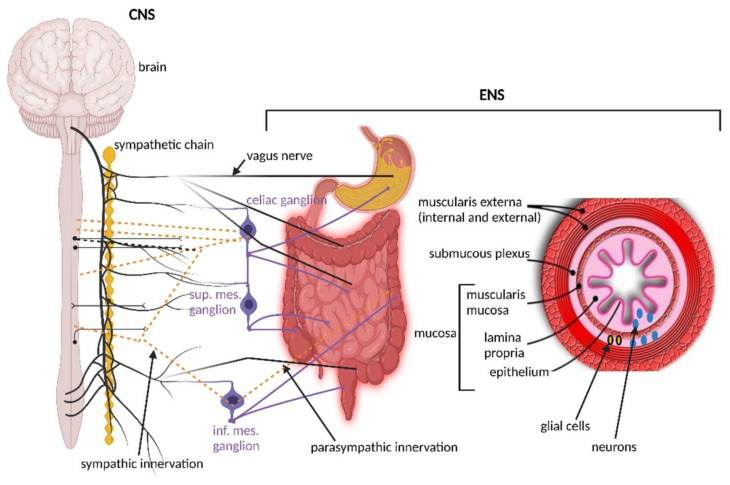
Although the vagus nerve (VN) is in contact with all layers of the gut wall, fibers do not cross the gut wall and are, thus, not in direct contact with gut microbiota. Signals reach the gut microbiota via 100 to 500 million neurons in the enteric nervous system (ENS) in the submucosa and myenteric plexi of the intestinal wall. The ENS in the small and large intestinal tract functions independently from the VN. This illustration was constructed using BioRender (https://biorender.com/, assessed on 15 July 2022).

## Data Availability

Not applicable.

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
