# Peer review of "Gut Bacteria and Neurotransmitters"

_microorganisms, 2022, doi:10.3390/microorganisms10091838_

Round 1

Reviewer 1 Report

 This article reviewed gut microbiome role in regulation of neurotransmitters. It’s an interesting topic and have a raised valid point and will be interesting for readers working related. Broadly the author demonstrated that gut microbiome together with food degradation, immune balance in the gut and regulation of enteroendocrine signalling pathway, its also communicate with the CNS through production of various secondary bile acids, short-chain fatty acids (SCFAs), glutamate (Glu), γ-aminobutyric acid (GABA), dopamine (DA), norepinephrine (NE), serotonin (5-HT) and histamine etc. Here the author explained how this various molecule production in the gut influence the host neurological process.

1. Check for some spelling mistakes and space in the text.

Author Response

Thank you for the comments.

All spelling and spaces has been checked and highlighted in text.

Yours sincerely

Prof LMT Dicks 

Reviewer 2 Report

Comments to the Author: I found this paper to be well written, and the reviewed content is going to be very beneficial to the community – thanks for taking the time to assemble and review these references. After your manuscript has been accepted for publication, I will be using/citing your paper in my own gut-brain and gut-liver axes-based research endeavors.

If it fits within the journal format, I would suggest including a list of all abbreviations used in the paper – there is a very large number of abbreviations used in this manuscript, and I found myself having to page back and forth through the manuscript to just confirm the metabolite, peptide, or receptor being discussed.

 Comments:

Abstract: Page 1, Line 7: Consider changing “food degradation” to “digestion of food.”

 Introduction:

Section 2: N/A

Section 3: N/A

Section 4: N/A

Section 5.2, Line 389: Consider changing “co-incides” to “coincides.”

Section 5.6 Lines 448-449: I think that there may be a small text error in the sentence that begins with “Although 5-HT is synthesized…” I have the general idea of what is written because of my review of the literature, but someone that is newer to the field may struggle with the concept being described here. Please consider rewriting this sentence to clarify the intended message.

Section 5.6, Line 461: There is a misspelling. Change “synthesise” to “synthesize.”

Section 5.6, Line 490: Please add the enzyme commission number for tryptophan monoxygenase (EC 1.13.13.3) and indole-3-acetamide hydrolase (EC 3.5.1.4). It would be useful to the interested reader if the EC numbers were listed for all enzymes mentioned throughout the article.

Section 5.6, Line 490: It would be good to add a brief discussion on how microbial-derived SCFAs promote serotonin production by enterochromaffin cells that line in the colon to Section 5.6. See Reference Reigstad, et al., FASEB J, 2015, 29(4), 1395-1403 as a good start for this new text.

Section 6, Lines 589-590: Please add the enzyme commission number for dopamine-beta-hydroxylase (EC 1.14.17.1). It would be useful to the interested reader if the EC numbers were listed for all enzymes mentioned throughout the article.

Section 6, Lines 597-598: There is a misspelling. Change “synthesised” to “synthesized.”

Section 6, Lines 632-633 & 635-636: The sentence on lines 632-633 almost exactly matches the sentence on lines 635-636, and the reference cited in both sentences is identical. It would be a good idea to remove the duplicate sentence.

Questions:

Page 3, Line 125: Should this be glandular tissue, gastric glands, or intestinal gland instead of glandular glands?

Author Response

Thank you for the comments.

Comments to the Author: I found this paper to be well written, and the reviewed content is going to be very beneficial to the community – thanks for taking the time to assemble and review these references. After your manuscript has been accepted for publication, I will be using/citing your paper in my own gut-brain and gut-liver axes-based research endeavors.

If it fits within the journal format, I would suggest including a list of all abbreviations used in the paper – there is a very large number of abbreviations used in this manuscript, and I found myself having to page back and forth through the manuscript to just confirm the metabolite, peptide, or receptor being discussed.

Answer: A list of abbreviations have been added on page 1.

AbstractPage 1, Line 7: Consider changing “food degradation” to “digestion of food.”

Answer: Changed to digestion of food, as suggested (line 7).

Introduction:

Section 2: N/A

Section 3: N/A

Section 4: N/A

Section 5.2, Line 389: Consider changing “co-incides” to “coincides.”

Answer: Changed to coincides (line 422).

Section 5.6 Lines 448-449: I think that there may be a small text error in the sentence that begins with “Although 5-HT is synthesized…” I have the general idea of what is written because of my review of the literature, but someone that is newer to the field may struggle with the concept being described here. Please consider rewriting this sentence to clarify the intended message.

Answer: The sentence has been changed to read “Although 5-HT is synthesized by neurons of the ENS, more than 90% of 5-HT is produced in the gut by ECCs [146]” (lines 481 and 482).  The original value of 60 was incorrect and has now been changed to 90%, as in reference 146.

Section 5.6, Line 461: There is a misspelling. Change “synthesise” to “synthesize.”

Answer: Changed to synthesize (line 494).

Section 5.6, Line 490: Please add the enzyme commission number for tryptophan monoxygenase (EC 1.13.13.3) and indole-3-acetamide hydrolase (EC 3.5.1.4). It would be useful to the interested reader if the EC numbers were listed for all enzymes mentioned throughout the article.

Answer:  The EC numbers, where available, have been added throughout (and highlighted in the text).

Section 5.6, Line 490: It would be good to add a brief discussion on how microbial-derived SCFAs promote serotonin production by enterochromaffin cells that line in the colon to Section 5.6. See Reference Reigstad, et al., FASEB J, 2015, 29(4), 1395-1403 as a good start for this new text.

Answer: A discussion has been added and the suggested reference, plus and additional reference included (see lines 498 to 507).  Reference numbers have been changed with the addition of these two references.

Section 6, Lines 589-590: Please add the enzyme commission number for dopamine-beta-hydroxylase (EC 1.14.17.1). It would be useful to the interested reader if the EC numbers were listed for all enzymes mentioned throughout the article.

Answer: This has been done (page 7), and throughout the text (changes have been highlighted).

Section 6, Lines 597-598: There is a misspelling. Change “synthesised” to “synthesized.”

Answer: Now corrected (line 639).

Section 6, Lines 632-633 & 635-636: The sentence on lines 632-633 almost exactly matches the sentence on lines 635-636, and the reference cited in both sentences is identical. It would be a good idea to remove the duplicate sentence.

Answer: The sentence has been deleted from line 677 and the sentence in line 674 has been kept.

Questions:

Page 3, Line 125: Should this be glandular tissue, gastric glands, or intestinal gland instead of glandular glands?

Answer: Changed to “glandular tissue” (line 155).

Yours sincerely

Prof LMT Dicks